# Upfront or Deferred Autologous Stem Cell Transplantation for Newly Diagnosed Multiple Myeloma in the Era of Triplet and Quadruplet Induction and Minimal Residual Disease/Risk-Adapted Therapy

**DOI:** 10.3390/cancers15245709

**Published:** 2023-12-05

**Authors:** Clifton C. Mo, Monique A. Hartley-Brown, Shonali Midha, Paul G. Richardson

**Affiliations:** Department of Medical Oncology, Dana-Farber Cancer Institute, Jerome Lipper Center for Multiple Myeloma Research, Harvard Medical School, 450 Brookline Avenue, Dana 1B02, Boston, MA 02115, USA; clifton_mo@dfci.harvard.edu (C.C.M.); moniquea_hartley-brown@dfci.harvard.edu (M.A.H.-B.); shonali_midha@dfci.harvard.edu (S.M.)

**Keywords:** autologous stem cell transplantation, genotoxicity, high-dose melphalan, minimal residual disease, multiple myeloma, newly diagnosed, quadruplets, transplant-eligible, treatment personalization, triplets

## Abstract

**Simple Summary:**

Patients who are diagnosed with multiple myeloma are given an initial sequence of treatments that usually, for those who are young and fit enough, includes high-dose melphalan followed by autologous stem cell transplantation. This has contributed to the improvement in survival seen over the past 30 years. However, high-dose melphalan has significant limitations, including short-term side effects and longer-term issues such as an increased risk of developing secondary hematologic malignancies including leukemia. There are now numerous highly efficacious combination regimens for initial treatment that result in increasingly large proportions of patients achieving deep responses with no evidence of minimal residual disease. Moreover, large, randomized studies using these regimens have shown no benefit in overall survival after receiving high-dose melphalan with stem cell transplantation. There is thus a growing rationale for selected eligible patients to defer receiving high-dose melphalan and stem cell transplantation until potentially needed in a subsequent line of treatment.

**Abstract:**

The standards of care for the initial treatment of patients with newly diagnosed multiple myeloma (NDMM) who are eligible for high-dose melphalan and autologous stem cell transplantation (HDM-ASCT) include highly active triplet and quadruplet regimens based on proteasome inhibitors, immunomodulatory drugs, and monoclonal antibodies. These regimens are resulting in improved outcomes and increasingly high rates of minimal residual disease (MRD)-negative responses without HDM-ASCT as part of the upfront therapy. Furthermore, recent randomized studies have shown that, while transplant-based approaches as a frontline therapy result in significantly longer progression-free survival compared to non-transplant approaches, this has not translated into an overall survival benefit. Given these developments, and in the context of the treatment burden of undergoing HDM-ASCT, in addition to the acute toxicities and long-term sequelae of HDM, which are associated with the genotoxicity of melphalan, there is an increasing rationale for considering deferring upfront HDM-ASCT in select transplant-eligible patients and saving it as a treatment option for later salvage therapy. Here, we review the latest clinical trial data on upfront or deferred HDM-ASCT and on the activity of quadruplet induction regimens, including rates of MRD-negative responses, and summarize emerging treatment approaches in the upfront setting such as the use of MRD-directed therapy and alternatives to HDM-ASCT.

## 1. Introduction

Multiple myeloma (MM) is the second most common individual hematologic malignancy [1,2], with an estimated global incidence of almost 180,000 new cases and 120,000 deaths in 2020, comprising approximately 1% of the global cancer burden [1]. The disease more commonly affects males (~56% of cases) and is generally a disease of the elderly, with the median age at diagnosis in the United States being 69 years [3]. MM exhibits substantial heterogeneity at diagnosis and throughout the disease course associated with multiple disease-related and patient-related characteristics including disease stage [4,5], cytogenetic abnormalities [6], age, and frailty [7,8], providing the context for the drive to develop personalized treatment approaches [9,10,11]. Overall survival (OS) has increased markedly over the past four decades, with the 5-year survival rate in the United States more than doubling to 59.8% [3] and the median OS in younger, fitter patients reaching approximately 10 years [12]. This is associated with the introduction and widespread adoption of high-dose melphalan plus autologous stem cell transplantation (HDM-ASCT) as a frontline therapy in eligible patients and, more importantly, the more recent development and use of numerous highly active novel agents and regimens throughout the disease course in multiple lines of therapy [13,14]. Such progress is rapid and ongoing, as evidenced by the recent approvals by the United States Food and Drug Administration (FDA) in 2021–2023 of the chimeric antigen receptor (CAR) T cell therapies idecabtagene vicleucel (ide-cel) and ciltacabtagene autoleucel (cilta-cel) [15,16], and of the bispecific antibodies teclistamab, elranatamab, and talquetamab [17,18,19].

### Current Treatment of Newly Diagnosed MM (NDMM) and the Role of HDM-ASCT

The current standards of care for the treatment of NDMM are based on three classes of agents: the proteasome inhibitors (PIs; bortezomib, carfilzomib, ixazomib), the immunomodulatory drugs (lenalidomide, pomalidomide, thalidomide), and the monoclonal antibodies (mAbs; anti-CD38 mAbs daratumumab, isatuximab; anti-SLAMF7 mAb elotuzumab) [13,20]. These agents are administered in combination—typically with dexamethasone—as triplet and, increasingly, quadruplet induction therapies, and as single-agent or doublet maintenance regimens as part of frontline therapy [13,20]. Post-induction consolidation therapy depends on a patient’s eligibility for HDM-ASCT, which remains the standard approach for patients aged ≤65–70 years without contraindicating comorbidities [13,20].

The use of HDM-ASCT as a standard in transplant-eligible patients was initially established based on randomized trials versus chemotherapy in the era prior to novel agents, in which transplant resulted in improved progression-free survival (PFS) and OS [21,22]. More recently, large phase 2 and phase 3 studies incorporating triplet novel-agent induction, with or without consolidation, and maintenance therapy have further demonstrated that the addition of HDM-ASCT confers a highly significant PFS benefit [23,24,25,26,27,28]. For example, the DETERMINATION phase 3 trial showed that the addition of HDM-ASCT to lenalidomide-bortezomib-dexamethasone (RVd) induction, plus lenalidomide maintenance to progression, resulted in a nearly 2-year median PFS benefit (67.5 vs. 46.2 months) and a 35% reduction in the risk of progression (RVd-alone vs. RVd + ASCT hazard ratio [HR] 1.53) [24]. The Intergroupe Francophone du Myélome (IFM) 2009 phase 3 trial, which had a similar design but administered lenalidomide maintenance for 1 year only, also showed a substantial PFS benefit with the addition of a transplant (median PFS 47.3 vs. 35.0 months, HR 1.43), although this was markedly less than the duration of disease control seen in both arms of DETERMINATION [23].

However, the importance and feasibility of the personalization of therapy is growing [9,10,11]. In the context of increasingly active quadruplet induction regimens [29,30,31,32] and our growing understanding of the mutagenic effects of melphalan [24,33,34,35,36], as well as no OS benefit having been demonstrated with HDM-ASCT in recent studies [23,24,25,26,37], the role of HDM-ASCT as a standard approach for all-comers in transplant-eligible NDMM is being challenged. Indeed, several treatment guidelines and recommendations are including deferred HDM-ASCT as a possible option for select patients in the frontline setting (Table 1). In this review, we explore the drivers behind the increasing use of deferred HDM-ASCT and a more tailored approach to therapy, as well as evaluating future treatment approaches for NDMM based on minimal residual disease (MRD) status and the incorporation of the next generation of novel immune-based therapies.

## 2. The Challenge of Comparing Upfront Versus Deferred HDM-ASCT

It is important to acknowledge that directly comparing outcomes with upfront or deferred HDM-ASCT is challenging due to multiple potential confounders. For example, there is an inherent immortal time bias towards patients who receive deferred HDM-ASCT as a second-line therapy, because these patients must have already received frontline therapy and must still be young and fit enough to undergo HDM-ASCT as part of their second-line treatment [42]. Early analyses suggested that there were no differences in OS between the two approaches, and this may have been due, in part, to such potential bias [42,43,44]. However, there is the potential for bias in the opposite direction too if the group receiving deferred ASCT largely includes patients with an earlier need for second-line therapy following failure of their front-line regimen, i.e., those with more aggressive relapses. Moreover, in the context of the rapidly expanding range of highly active treatment options for relapsed/refractory MM (RRMM), a PFS benefit with upfront HDM-ASCT versus a non-transplant approach could result in a delayed need for second-line therapy, during which time additional novel, active treatment options might be approved in this setting.

Furthermore, for such comparisons, trials with a lengthy follow-up are required in order to evaluate outcomes through both first- and second-line therapy, which may be substantial for transplant-eligible patients in the modern era [12]. The IFM 2009 and DETERMINATION trials provide valuable data in this regard, as both trials recommended HDM-ASCT as second-line therapy following RVd alone [24,25]. In IFM 2009, with a median follow-up of 7.5 years, 262/350 (74.9%) and 217/350 (62.0%) patients on the RVd-alone and RVd + ASCT arms, respectively, required second-line therapy, with 201/262 (76.7%) versus 49/217 (22.6%) of them having received ASCT as part of that treatment. As in earlier studies of early versus later transplant, no difference in OS (8-year rate: 60.2% vs. 62.2%) was seen between arms [23], suggesting that deferred ASCT as part of the second-line therapy represents a reasonable clinical option. This is supported by several studies demonstrating substantial efficacy with HDM-ASCT in the RRMM setting [14,45,46]. Interestingly, however, in DETERMINATION no difference in OS (5-year rate: 79.2% vs. 80.7%) was seen between RVd alone and RVd + ASCT after a median follow-up of almost 6.5 years [24], despite only 78/279 (28.0%) RVd-alone patients who had discontinued protocol therapy having received subsequent HDM-ASCT. These findings, in the context of the large PFS benefit with RVd + ASCT, suggest the possibility of competing risk impacting OS.

## 3. The Rationale for Deferring HDM-ASCT

In addition to the absence of an OS benefit to date with RVd + ASCT versus RVd-alone in the IFM 2009 and DETERMINATION trials, several other factors contribute to the rationale for deferring HDM-ASCT and the design of treatment protocols by which to do so.

### 3.1. Acute Adverse Impacts of High-Dose Melphalan

HDM-ASCT may be associated with an increased risk of certain acute toxicities, which are important to bear in mind when considering upfront or deferred HDM-ASCT approaches. While treatment-related mortality rates are now low [24,25,47], HDM nevertheless results in the prolonged suppression of bone marrow function and significantly higher rates of severe hematologic toxicities than induction therapy alone, notably neutropenia [48], along with an associated increased risk of infection. For example, in DETERMINATION, the overall rate of any hematologic adverse events (AEs) over the course of treatment was 89.9% on the RVd + ASCT arm compared with 60.5% with RVd alone, with rates of febrile neutropenia (9.0% vs. 4.2%) and pneumonia (9.0% vs. 5.0%) also higher with RVd + ASCT [24]. Similarly, the incidence of grade 3/4 neutropenia (92% vs. 47%) and grade 3/4 infections (20% vs. 9%) was higher with RVd + ASCT versus RVd alone in the IFM 2009 study [25].

HDM is also associated with gastrointestinal disorders, including high rates of diarrhea, nausea, and vomiting, as well as esophageal, gastric, and colonic mucosal injury [49,50,51]. Grade 3/4 gastrointestinal AEs were reported in 28% versus 7% of patients receiving RVd + ASCT versus RVd alone in IFM 2009 [25], and the rate of treatment-related gastrointestinal AEs was similarly higher (19% vs. 8%) in DETERMINATION [24]. Moreover, HDM is specifically also associated with oral mucositis [52], with no such AEs reported in the RVd-alone arm of DETERMINATION [24].

The acute toxicities and the treatment burden of HDM can have a marked adverse impact on patients’ quality of life (QoL) associated with the transplant procedure, although QoL measures subsequently recover and may further improve compared to a pretreatment baseline [53]. Transient but clinically meaningful decreases in QoL were seen in the RVd + ASCT arm in both DETERMINATION [24] and IFM 2009 [54], with subsequent recovery during medium-term follow-up. These included transient worsening of the Global Health Status, Physical Functioning, and Role Functioning domain scores of the European Organisation for Research and Treatment of Cancer (EORTC) QoL Questionnaire–Core 30 module (QLQ-C30) and of the Side Effects score of the EORTC-QLQ Myeloma-specific module (MY20) [24,54].

### 3.2. Long-Term Sequelae of HDM

The potential for long-term sequelae of HDM associated with the genotoxicity of melphalan [33] is an important consideration when potentially deferring transplant. MM evolution and progression has been shown to be characterized by a number of mutational processes, including the melphalan-specific single-base substitution (SBS)-MM1 mutational signature [35,55,56,57], and it has been demonstrated that HDM exposure results in a significantly increased overall mutational burden in residual MM at the time of relapse [34]. Consequently, there is an increased risk of second primary malignancies (SPMs) in MM patients following HDM-ASCT—data from the United States National Cancer Institute Surveillance, Epidemiology, and End Results (SEER) Program showed that the risk of acute myeloid leukemia (AML) and myelodysplastic syndromes (MDS) in MM patients is 5–10 times the background rate in the general population, while an analysis of data from the Center for International Blood and Marrow Transplant Research (CIBMTR) demonstrated relative risks of 10–50 for AML and ~100 for MDS in a cohort of MM patients who received HDM-ASCT [36]. This is supported by recent clinical trial data from DETERMINATION, which showed a significantly higher rate of secondary AML/MDS in the RVd + ASCT versus RVd-alone arm (2.7% vs. 0%, *p* = 0.002) in the context of continuous lenalidomide maintenance post-ASCT/induction [24].

As highlighted in a recently published analysis of CIBMTR registry data, the risk of SPMs is an important survivorship issue for MM patients who underwent HDM-ASCT followed by lenalidomide maintenance [58]. On multivariate analysis, the CIBMTR data showed that having any SPM was associated with significantly shorter PFS (HR 2.62) and OS (HR 5.01), an association that appeared stronger for hematologic SPMs (PFS HR 3.85, OS HR 8.13) [58]; thus, SPMs may be a competing risk contributing to the lack of OS benefit with transplant-based versus non-transplant-based front-line therapy [23,24]. Similar findings were reported from an analysis of MM patients in the Netherlands Cancer Registry, in which the development of an SPM was associated with a greater mortality risk [59]; notably, these data showed a significant increase in the risk of AML/MDS SPMs over time, from 1994–2000 to 2001–2007 and 2008–2013, coinciding with the increasing use of HDM-ASCT for MM [59].

Other long-term survivorship issues post-HDM may include an increased risk of hematologic complications with subsequent therapies—an analysis reported at the 2022 Annual Meeting of the American Society of Hematology (ASH) showed that a prior history of ≥1 HDM-ASCT was correlated with poor hematologic recovery after CAR T cell therapy, which, in turn, showed a possible association with subsequent MDS [60]. Furthermore, an analysis of 630 MM patients who had survived for >2 years following HDM-ASCT showed that, compared with sibling controls, MM patients treated with HDM-ASCT had 40% greater odds of developing severe and/or life-threatening chronic health conditions (CHCs), with a 10-year cumulative incidence of 58% [61]; these included venous thromboembolism, subsequent neoplasms, and, cataracts. A recent analysis of long-term survivors at a median of 4 years post-transplant also found that around a third of patients had clinically significant distress (per the Cancer- and Treatment-Related Distress [CTXD] instrument), most notably in the domain of ‘Health Burden’ [62]. These findings indicate the need for long-term monitoring to help manage subsequent morbidity and complications post-transplantation.

### 3.3. Personalization of Treatment and Patient Preferences

The current treatment armamentarium for MM contains a large number of highly active salvage therapy options and so, in the modern era and in regions with access to these multiple options, an upfront PFS benefit may no longer translate into an OS benefit due to the activity of subsequent lines of treatment [24,47,63]. In this context, real-world aspects of care, including patients’ preferences for treatment and their personal circumstances, are important when considering frontline therapy choices. For example, HDM-ASCT represents an intensive treatment approach that can place a substantial burden on patients in terms of the need for a period of hospitalization and recovery that will typically last a number of weeks—this may not be feasible for some patients, depending on their circumstances (e.g., dependents, work), and so they may prefer to select a more convenient and tolerable non-transplant option with less impact on their daily lives [11].

Furthermore, given the range of different options, and the goal of the personalization of treatment, it is important to explore the benefit of treatment modalities in patients with specific demographic, biologic, or disease-related characteristics. Indeed, subgroup data from the DETERMINATION trial support the idea that ‘one size does not fit all’—with RVd alone versus RVd + ASCT, HRs for PFS ranging from 0.96 to 3.40 were seen across patient subgroups defined by race, body mass index (BMI), disease stage, and cytogenetics [24]. Preliminary analyses have shown that there may be a differential impact of prognostic factors between the non-transplant RVd-alone and RVd + ASCT treatment approaches [64], suggesting that some factors, such as race, BMI, disease stage, and cytogenetics, may be associated with a greater or lesser benefit from these therapies. For example, with RVd alone versus RVd + ASCT in DETERMINATION, median PFS was 44.3 versus 67.2 months (HR 1.67) in white patients but was not reached versus 61.4 months (HR 1.07) in African American patients [24]. Similarly, in patients with a BMI of <25, median PFS was 33.6 months versus not reached (HR 2.60) with RVd alone versus RVd + ASCT, but the magnitude of benefit was notably lower in patients with a BMI of 25 to <30 (median PFS 52.3 versus 64.3 months, HR 1.24) or ≥30 (median PFS 45.8 versus 64.4 months, HR 1.41) [24]. PFS and OS benefits with RVd + ASCT also varied according to cytogenetics. Among patients with high-risk cytogenetic abnormalities [del17p, t (4;14), t (14;16)] receiving RVd alone versus RVd + ASCT, median PFS was 17.1 versus 55.5 months (HR 1.99) and 5-year OS was 54.3% versus 63.4% (HR 1.25), suggesting an emerging OS benefit; these benefits appeared particularly pronounced in patients with t (4;14), with PFS and OS HRs of 2.72 and 1.39, respectively, but less so in patients with del17p (PFS and OS HRs of 1.44 and 1.03, respectively). By contrast, in patients with standard-risk cytogenetics, median PFS was 53.2 versus 82.3 months (HR 1.38) and 5-year OS was almost identical at 86.2% versus 86.0% (HR 0.99) [24]. Further research into patient and disease characteristics and other biomarkers with an association with outcome is warranted to inform tailored treatment approaches in the future.

### 3.4. Depth of Response and Rate of MRD Negativity with Quadruplet Induction

One prognostic marker that is already emerging as a potentially important way of guiding treatment escalation or de-escalation, including the use of deferred HDM-ASCT, is MRD status. Achieving MRD negativity has been demonstrated to be one of the key prognostic factors for long-term PFS and OS in patients with NDMM and a potential surrogate for outcomes [65,66,67,68,69]. Moreover, achieving sustained MRD negativity is highly correlated with prolonged PFS and OS [70,71,72,73,74] and is associated with better prognosis than simply achieving MRD-negative status; for example, an analysis of 23 patients with NDMM who were receiving lenalidomide maintenance following HDM-ASCT (n = 10) or induction therapy (n = 13) showed that patients who lost or were unable to obtain an MRD-negative status during the first year of maintenance were 14 times more likely to have disease progression than those with a sustained MRD-negative status for 1 year [70]. Notably, the immune milieu in patients with a sustained MRD-negative status who did not undergo HDM-ASCT was correlated with that in the bone marrow from healthy (non-MM) individuals [70]. The threshold for defining MRD-negative status (e.g., 10^−5^ or 10^−6^) also impacts its prognostic value, with MRD negativity resulting from more sensitive evaluation (i.e., 10^−6^ vs. 10^−5^) having greater prognostic value [67,75]; therefore, achieving MRD negativity at 10^−5^ sensitivity may not be optimal compared with 10^−6^.

Importantly, the prognostic value of MRD-negative status is independent of treatment [65,66,67,68], with a similar PFS seen in MRD-negative patients receiving transplant or non-transplant approaches [24,28]. For example, in preliminary data from DETERMINATION, among the 39.8% and 54.4% of patients in the RVd-alone and RVd + ASCT arms who were evaluated for MRD status at the start of maintenance and were MRD-negative, the 5-year PFS rates were 59.2% and 53.5%, respectively [24]. Thus, patients who achieve an MRD-negative response to treatment prior to planned transplant may be appropriate for a deferred HDM-ASCT approach.

Rates of MRD negativity have been increasing substantially with the use of modern triplet and, notably, quadruplet induction and consolidation regimens, either in conjunction with HDM-ASCT or alone. As shown in Table 2, MRD rates of up to 81% have been achieved with frontline treatment comprising quadruplet regimens plus HDM-ASCT [76], with MRD rates increasing throughout the course of frontline therapy from induction to transplant to consolidation to maintenance [77]. However, similarly high rates have also been seen with non-transplant approaches, with an MRD-negative response rate of 71% reported in the MANHATTAN trial of daratumumab plus carfilzomib-lenalidomide-dexamethasone [30]. Collectively, these data suggest that a substantial proportion of transplant-eligible patients may be able to defer HDM-ASCT if they achieve sustained MRD negativity with their induction therapy.

## 4. Discussion and Future Perspectives

In the context of the ongoing development of highly active quadruplet frontline regimens, and the increasingly high rates of MRD negativity being achieved, prospective clinical trials incorporating MRD-adapted therapeutic approaches for NDMM are underway to evaluate the feasibility of deferred HDM-ASCT in patients who achieve MRD-negative status following induction therapy. For example, the phase 3 MIDAS trial (NCT04934475; IFM 2020-02) is evaluating MRD status in patients following six cycles of quadruplet induction therapy with isatuximab, carfilzomib, lenalidomide, and dexamethasone (Isa-KRd); those who achieve MRD negativity at a sensitivity of 10^−5^ are then randomized to receive either a further six cycles of Isa-KRd or another two cycles of Isa-KRd plus HDM-ASCT, both followed by lenalidomide maintenance for 3 years. The primary objective of the study is to compare the rates of MRD negativity at 10^−6^ sensitivity achieved with these two approaches at various time-points during treatment.

Similarly, in the phase 2 ADVANCE study (NCT04268498), patients are being randomized to receive daratumumab plus KRd (Dara-KRd) or KRd alone for eight cycles, followed by the option for ASCT in MRD-positive patients or proceeding straight to lenalidomide maintenance for ≥2 years in MRD-negative patients. The MASTER-2 study (NCT05231629) is also stratifying patients by MRD status following daratumumab-based quadruplet induction—in this case, six cycles of daratumumab, bortezomib, lenalidomide, and dexamethasone (Dara-RVd)—with MRD-negative patients then being randomized to receive a further three cycles of Dara-RVd or HDM-ASCT, followed by maintenance therapy with daratumumab plus lenalidomide. Meanwhile, the PERSEUS phase 3 study (NCT03710603) of Dara-RVd versus RVd is using a different approach, with patients on the Dara-RVd arm receiving Dara-lenalidomide maintenance and stopping the Dara component upon achieving sustained MRD negativity (with the opportunity to then restart Dara upon recurrence of MRD). Results from these studies and others are awaited with interest to see whether outcomes are similar with a deferred HDM-ASCT approach in MRD-negative NDMM.

Furthermore, novel consolidation therapies are being explored as potential alternatives to HDM-ASCT. For example, the phase 3 CARTITUDE-6 trial (NCT05257083) [84] is comparing the use of the CAR T cell therapy ciltacabtagene autoleucel with HDM-ASCT in conjunction with six cycles of Dara-RVd and lenalidomide maintenance for 2 years, to determine which approach is superior in terms of PFS and sustained MRD-negative CR rate. Meanwhile, a number of studies are being developed to explore the use of bispecific antibodies such as teclistamab, talquetamab, and elranatamab as alternatives to HDM-ASCT—for example, the phase 2 GEM-TECTAL study (NCT05849610) is investigating intensification therapy with teclistamab plus daratumumab, with or without subsequent talquetamab plus daratumumab, in patients with high-risk NDMM.

With such studies in progress, it is likely that the treatment algorithm for patients with NDMM, particularly those who are transplant-eligible, will evolve and become more complex in the future. As shown in Figure 1, pending data from current and planned studies there may be multiple available treatment pathways for transplant-eligible patients including the use of deferred HDM-ASCT or its replacement with alternative intensification.

## 5. Conclusions

Based on the latest data from randomized clinical trials in patients with NDMM and given the context of there being numerous highly active novel treatment options and the need for a personalized approach to treatment, incorporating a longer-term strategic view of patient outcome, especially in the era of immune therapy, deferred HDM-ASCT is emerging as a standard-of-care approach for select transplant-eligible patients. With no OS benefit being seen in multiple randomized studies [23,24,25,26,37] of transplant versus non-transplant approaches for NDMM, the increasingly high rates of MRD negativity achieved with quadruplet induction regimens, and the development of MRD- and risk-adapted treatment approaches, NDMM therapy is evolving from a one-size-fits-all approach of upfront HDM-ASCT to a response-adapted, risk-modified and strategic therapeutic paradigm, with the aim of further improving outcomes for patients and enhancing quality of life.

## Figures and Tables

**Figure 1 cancers-15-05709-f001:**
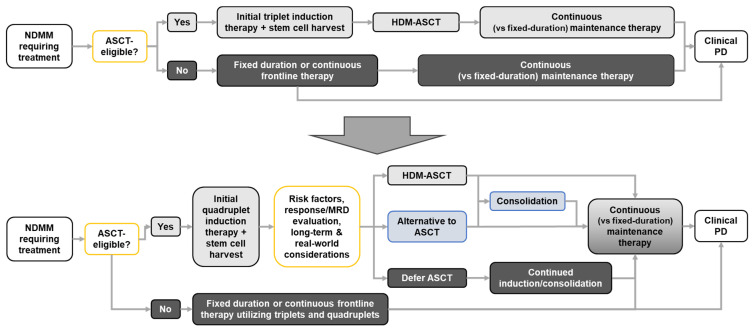
Potential future evolution of the treatment algorithm for patients with NDMM.

**Table 1 cancers-15-05709-t001:** Recent treatment guidelines and recommendations for NDMM including early or deferred HDM-ASCT.

Publication	Year Published	Early HDM-ASCT	Deferred HDM-ASCT
EHA-ESMO Clinical Practice Guidelines [20]	2021	“For patients <70 years without comorbidities, induction therapy followed by HDM and ASCT is the recommended treatment”	Not included
BSH/Myeloma UK guidelines [38]	2021	“Recommended for younger, fitter patients”	“Lack of OS benefit … likely to be largely due to the use of delayed ASCT … supports the use of deferred ASCT as a clinical option… [the fact that] patients in the non-ASCT arm of the IFM 2009 study were unable to receive ASCT at relapse due to disease refractoriness reinforces the benefit of upfront ASCT where feasible”
ASTCT Clinical Practice Recommendations [39]	2022	“The panel recommends early autologous transplantation as a consolidation therapy in eligible, newly diagnosed myeloma patients after 4–6 cycles of induction”	“The panel recommends mobilization and storage of peripheral blood stem cells in newly diagnosed myeloma patients not undergoing autologous transplantation after first line of therapy for future use as a treatment at first relapse”
Rajkumar, update on diagnosis, risk-stratification and management [40]	2022	“ASCT should be considered in all eligible patients”	“In standard-risk patients responding well to therapy, ASCT can be delayed until first relapse provided stem cells are harvested early in the disease course”
mSMART guidelines [41]	2023	Preferred for standard-risk patients [t (11;14), t (6;14), trisomies], recommended for high-risk patients	An option for standard-risk patients

ASCT, autologous stem cell transplantation; ASTCT, American Society for Transplantation and Cellular Therapy; BSH, British Society of Haematology; EHA, European Hematology Association; ESMO, European Society for Medical Oncology; HDM, high-dose melphalan; mSMART, Stratification for Myeloma and Risk-adapted Therapy; NDMM, newly diagnosed multiple myeloma.

**Table 2 cancers-15-05709-t002:** High rates of MRD negativity reported with quadruplet regimens in NDMM, with or without HDM-ASCT.

Study	Induction/Consolidation	MRD-Negative Rate	Outcomes
GMMG-HD7 [78]	Induction: 3 × Isa-RVd 6-week cycles	Post-induction: 50%	NR
IFM 2018-01 [79]	Induction: 6 × Dara-IRd (3-week cycles) ASCT Consolidation: 4 × Dara-IRd (4-week cycles)	10^−5^/10^−6^ sensitivity Post-induction: 28%/6% Post-ASCT: 34%/29% Post-consolidation: 51%/40%	2-year PFS: 95.2%
IFM 2018-04 [80] Patients with high-risk cytogenetics	Induction: 6 × Dara-KRd ASCT Consolidation: 4 × Dara-KRd (4-week cycles)	Post-induction: 62% (10^−5^)	18-month PFS: 92% 18-month OS: 96%
GMMG-CONCEPT [81] High-risk MM	Induction: 6 (TE)/8 (TIE) × Isa-KRd ASCT (TE) Consolidation: 4 × Isa-KRd (4-week cycles)	Post-consolidation (10^−5^): TE: 68%; TIE: 54%	NR
CASSIOPEIA [31]	Induction: 4 × Dara-VTd ASCT Consolidation: 2 × Dara-VTd (4-week cycles)	100 days post-ASCT (10^−5^): 64%	18-month PFS: 93%
GRIFFIN [77]	Induction: 4 × Dara-RVd ASCT Consolidation: 2 × Dara-RVd Maintenance: DR	10^−5^/10^−6^ sensitivity Post-induction: 22%/1% Post-consolidation: 50%/11% Post-1-year-maintenance: 59%/21% End of study: 64%/36%	4-year PFS: 87.2% 4-year OS: 92.7%
MANHATTAN [30]	Induction: 8 × Dara-KRd (4-week cycles) No ASCT	10^−5^: 71%	1-year PFS: 98% 1-year OS: 100%
MASTER [76]	Induction: 4 × Dara-KRd (4-week cycles) ASCT Consolidation: 0, 4, or 8 × Dara-KRd	Post-consolidation (10^−5^/10^−6^): 81%/71%	0/1/2 HRCA 3-year PFS: 91%/87%/51% 3-year OS: 96%/91%/75%
NCT02969837 [82]	Induction: 12—24 × Elo-KRd (4-week cycles) No ASCT	10^−5^/10^−6^ sensitivity After 8 cycles: 63%/51% Best response: 70%/60%	3-year PFS: 72% 3-year OS: 78%
SKylaRk [83]	Induction: 4 × Isa-KRd (4-week cycles) Optional ASCT If ASCT deferred: 4 × Isa-KRd (4-week cycles)	10^−5^/10^−6^ sensitivity Post-cycle 4 (n = 28): 43%/32%	1-year PFS: 97.9% 1-year OS: 97.9%

ASCT, autologous stem cell transplantation; d, dexamethasone; Dara, daratumumab; Elo, elotuzumab; HRCA, high-risk cytogenetic abnormalities; I, ixazomib; Isa, isatuximab; K, carfilzomib; MM, multiple myeloma; MRD, minimal residual disease; NR, not reported; OS, overall survival; PFS, progression-free survival; R, lenalidomide; T, thalidomide; TE, transplant-eligible; TIE, transplant-ineligible; V, bortezomib.

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
