# Peer review of "Upfront or Deferred Autologous Stem Cell Transplantation for Newly Diagnosed Multiple Myeloma in the Era of Triplet and Quadruplet Induction and Minimal Residual Disease/Risk-Adapted Therapy"

_cancers, 2023, doi:10.3390/cancers15245709_

Round 1

Reviewer 1 Report

Comments and Suggestions for Authors

It is a review of data that in MM should be considered in the planning of future studies. The major point is that the use of autologous hematopoietic transplants could safely be deferred. The arguments sustaining this opinion are very well gathered and analysed.

In my review, I did not find any issue needing clarification.

- The main question addressed by the research is the rationale to treat multiple myeloma patients avoiding high dose chemotherapy.   - It is an important topic and relevant to the field.   - Compared with other published material, this subject area adds that these considerations are not frequently presented and organized in a comprehensive way.   - No further specific improvement can be suggested.   - The argument is deeply discussed.   - References are appropriate and updated   - No specific comment on the figures and tables.

Author Response

We thank the reviewer for their kind feedback.

Reviewer 2 Report

Comments and Suggestions for Authors

The manuscript by Mo et al. represents a review article summarizing the recent advances in the upfront or deferred autologous stem cell transplantation for newly diagnosed multiple myeloma in the era of triplet and quadruplet induction and minimal residual disease/risk-adapted therapy. While this comprehensive analysis is timely, the manuscript is well written, well illustrated, and easy to follow.  Based on the analysis of the literature on randomized clinical trials in patients with NDMM, the authors summarize the current state of the field and outline the prospects for the future development of a strategic therapeutic paradigm with the goal of further improving patient outcomes and quality of life. I think the manuscript is ready to be published as it stands.

Author Response

(The authors gave the same response as above.)

Reviewer 3 Report

Comments and Suggestions for Authors

his critical review is very updated. More than 75% of the references are of the last four years (2020-2023): 17 from 2023, 20 from 2022, 15 from 2021, and 13 from 2020. The authors analyze the latest clinical trial data on upfront or deferred high-dose melphalan (HDM) and autologous stem cell transplantation (ASCT) and on the activity of quadruplet induction regimens, including rates of minimal residual disease (MRD)-negative responses, and summarize emerging treatment approaches in the upfront setting such as the use of MRD-directed therapy and alternatives to HDM-ASCT.

This review contains two tables and one figure. Table 1 describes the recent treatment guidelines and recommendations for newly diagnosed myeloma multiple including early or deferred high-dose melphalan and autologous stem cell transplantation. It includes guidelines and recommendations published recently (2021-2023).

Table 2 analyzes ten studies (7 of 2022, 2 of 2021 and 1 of 2019) of the high rates of minimal residual disease (MRD) negativity reported with quadruplet regimens in newly diagnosed myeloma multiple (NDMM), with or without high-dose melphalan and autologous stem cell transplantation (HDM-ASCT).

Figure 1 analyzes the treatment strategies that could be followed in patients with NDMM requiring treatment depending if they are eligible or not for ASCT.

They conclude that deferred HDM-ASCT may be the best approach for tho patients that are eligible for ASCT.

Minor:

RRMM and IFM should be defined.

Table 1: Foot note: NDMM should be defined.

Author Response

We thank the reviewer for their kind feedback. The abbreviations have been defined as requested on lines 105 (NDMM), 118 (RRMM), and 123-124 (IFM).